# Using a Tandem Flight Configuration between Sentinel-6 and Jason-3 to Compare SAR and Conventional Altimeters in Sea Surface Signatures of Internal Solitary Waves

Jorge M. Magalhaes [1,2], Ian G. Lapa [2], Adriana M. Santos-Ferreira [1,2], José C. B. da Silva [2,3,*], Fanny Piras [4], Thomas Moreau [4], Samira Amraoui [4], Marcello Passaro [5], Christian Schwatke [5], Michael Hart-Davis [5], Claire Maraldi [6] and Craig Donlon [7]

1   Interdisciplinary Centre of Marine and Environmental Research (CIIMAR), Terminal de Cruzeiros de Leixões, Av. General Norton de Matos, s/n, 4450-208 Matosinhos, Portugal
2   Department of Geoscience, Environment and Spatial Planning (DGAOT), Faculty of Sciences, University of Porto, Rua do Campo Alegre, s/n, 4169-007 Porto, Portugal
3   Instituto de Ciências da Terra, Polo Porto, Universidade do Porto, Rua do Campo Alegre 687, 4169-007 Porto, Portugal
4   Collect Localisation Satellites (CLS), 11 rue Hermès, Parc Technologique du Canal, 31520 Ramonville St. Agne, France
5   Deutshes Geodätisches Forschungsinstitut der Technischen Universität München (DGFI-TUM), Arcisstraße 21, 80333 Munich, Germany
6   Centre National d'Études Spatiales (CNES), 18 avenue Edouard Belin, 31401 Toulouse, France
7   European Space Agency/European Space Research and Technology Centre (ESA/ESTEC), Keplerlaan 1, 2201 AZ Noordwijk, The Netherlands
*   Correspondence: jdasilva@fc.up.pt

**Abstract:** Satellite altimetry has been providing a continuous record of ocean measurements with numerous applications across the entire range of ocean sciences. A reference orbit has been used since 1992 with TOPEX/Poseidon, which was repeated in the Jason missions, and in the newly launched Sentinel-6 Michael Freilich (in November 2020) to continually monitor the trends of sea level rise and other properties of the sea surface. These multidecadal missions have evolved alongside major technological advances, whose measurements are unified into a single data record owing to continuous intercalibration and validation efforts. However, the new Sentinel-6 provides synthetic aperture radar (SAR) processing, which improves the along-track resolution of conventional altimeters from a few kilometres (e.g., for Jason-3) to about 300 m. This means a major leap in sampling towards higher frequencies of the ocean spectrum, which inevitably means reconciling the assumption of a uniform Brown surface between the footprints of the larger kilometre-scale conventional altimetry and those of the finer-scale SAR altimetry. To explore this issue, this study uses the vantage point of the Sentinel-6/Jason-3 tandem phase to compare simultaneous sea surface signatures of large-scale Internal Solitary Waves (ISWs) between SAR and conventional altimetry. These waves can modulate the sea surface into arrayed sections of increased and decreased roughness with horizontal scales up to 10 km, which inflict sharp transitions between increased and decreased backscatter in the radar altimeters. It is found that Sentinel-6 can provide more detailed structures of ISWs in standard level-2 products, when compared with those from the conventional Jason-3 (similarly to previous results reported from the SAR altimeter from Sentinel-3). However, a new and striking feature is found when comparing the radar backscatter between Sentinel-6 and Jason-3, which are in opposite phases in the ISWs. These intriguing results are discussed in light of the intrinsically different acquisition geometries of SAR and conventional altimeters as well as possible implications thereof.

**Keywords:** Internal Solitary Waves (ISWs); conventional altimeter; delay-doppler altimeter; tandem phase; satellite altimetry reference orbit

## 1. Introduction

Satellite altimetry has been providing continuous monitoring of ocean dynamics for more than 30 years [1]. Its fundamental principle is quite simple and relies in measuring the time taken by a microwave radar pulse to travel from the satellite antenna to the sea surface and back, which translates to distance after a series of corrections have been accounted [2]. Nonetheless, over the last three decades, the contents of the returned radar echo have been found to provide additional information about the sea surface. For instance, level-2 altimetry products provide Sea Surface Height Anomalies (SSHAs), Significant Wave Heights (SWHs) and Normalized Radar Cross Sections (or radar backscatter, $\sigma_0$), which are routinely used in a wide range of applications ranging from operational oceanography to climatological studies (e.g., to gauge sea-level rise, for a detailed description of satellite altimetry applications see [3] and references therein).

Nowadays, motivations are being driven to survey near-shore coastal areas and inland water bodies (see e.g., [4]) which means improving ground resolution. In particular, conventional (pulse-limited) radar altimeters (with footprints of the order of a few kilometres) are being replaced with SAR altimeters. These new and more advanced altimetry sensors use Synthetic Aperture Radar (SAR) technology to improve their along-track resolution to a few hundred meters—which nonetheless retains the original across-track resolution of a few kilometres [5].

To ensure the continuity of a consistent and intercomparable global altimetry data record throughout all altimetry missions flown in these last 30 years (and those to come), new generations of satellite altimeters replace the previous missions in precisely the same orbit. This orbit is often referred to as the *satellite altimetry reference orbit*, and in their early stages both new and previous missions are placed in a tandem configuration with one satellite lagging the other (see e.g., [6–8]). This strategy allows the same ground-track to be repeated mission after mission, and hence ensures the continuity of the satellite altimetry record.

The recently launched Sentinel-6 Michael Freilich satellite [7] is no exception and its tandem phase with the previous altimetry reference mission (Jason-3, still in operation) was completed approximately between December 2020 and March 2022. However, Sentinel-6 provides for the first time the capability to acquire SAR altimeter data along the reference orbit of the previous multi-decadal altimetry missions. We note in passing that a SAR altimeter (SRAL) was already aboard Sentinel-3a and -3b (launched in 2016 and 2021), which was not in the same reference altimetry orbit. Therefore, in a way, the Sentinel-6 and Jason-3 tandem flight configuration is unique as it places for the first (and perhaps only) time a SAR and a Jason-class conventional radar altimeter in exactly the same orbit with one lagging the other by just 30 s. This means an opportunity to explore how previous conventional and new SAR altimeters sense the same phenomena under the same environmental conditions— namely, surface wind and wave fields are not expected to change in a timeframe of 30 s (except for swell waves as reported by Rieu et al. [9]).

Even so, all tandem missions are temporary (lasting no more than a few months), and hence offer only a limited amount of data. To overcome this issue, the new Sentinel-6 mission provides yet another novelty. Its Poseidon 4 altimeter can acquire simultaneously conventional and SAR altimetry data (sometimes referred to as Low- and High-Resolution Modes, respectively—i.e., LRM and HR, see e.g., [7]). In other words, the LRM in Sentinel-6 is meant as a heritage of the Jason-class altimeters (namely, Jason-3) which can be directly compared with its new SAR measurements throughout the entire mission in the altimetry reference orbit—and hence key for conventional/SAR altimetry intercalibration purposes. This adds to the motivations in the tandem configuration, since the ability of LRM data in Sentinel-6 to replicate that of Jason-3 can be verified directly (again with a small-time lag of just 30 s).

Nonetheless, while exploring the vantage points of this tandem configuration, it is important to recall that there are different instrumental and radiometric properties expected a priori between Sentinel-6 and Jason-3. But despite these differences, there is another potential source for inconsistencies between the measurements in these satellites. Despite

illuminating the same ground area with a time difference of just 30 s, the ground footprints in a conventional and a SAR altimeter are very different with one being much larger than the other. This is important because in the assumption of Brownian waveforms the properties of the illuminated sea surface are considered constant within the altimeter footprint (see e.g., [10,11]). In the real ocean that could mean that some phenomena can modulate the radar backscatter in scales smaller than the typical conventional altimeter footprint but larger than (or comparable to) the sharpened SAR along-track resolution—ultimately meaning that homogeneous surface roughness (i.e., a uniform Brown surface) could simultaneously be valid and invalid for SAR and conventional altimeters (respectively).

Therefore, to ensure robust and meaningful comparisons between the multidecadal legacy data from previous conventional altimetry missions and the upcoming SAR altimeters, it is important to understand how these distinct altimeters sense different phenomena and scales of the sea surface. Ideally, pursuing this goal would mean to intentionally break down the assumption of a constant sea surface backscatter within the altimeter footprint in spatial scales ranging between those of conventional and SAR altimeters, and assessing their different responses in comparison with some reference background.

Internal Solitary Waves (ISWs) in the ocean naturally provide this contrasting scenario in sea surface roughness. It has already been well-documented that these waves, which propagate along the ocean's pycnocline, can increase and decrease sea surface radar backscatter in scales of a few kilometres with quasi-instantaneous transitions between them—both in conventional altimeters [12] as well as in SAR altimeters [13–15].

Therefore, in this paper we focus on sea surface signatures of ISWs from the vantage point of a tandem configuration between Jason-3 and Sentinel-6, and use them to investigate how conventional and SAR altimeters compare while sensing sharp transitions in radar backscatter from the sea surface. The rest of the paper is organized as follows: Section 2 highlights the fundamental of conventional and SAR altimetry needed to interpret their sea surface signatures of ISWs; in Sections 3 and 4 we present and discussed our results; and the paper concludes in Section 5 with a summary of our findings.

## 2. ISW Observations in a Tandem Configuration between Sentinel-6 and Jason-3

The details of satellite altimetry needed to interpret ISW sea surface signatures have already been described in the literature and the reader is referred to the studies presented in [12–16] and references therein. Nonetheless, it is important to recall and discuss some fundamental differences between SAR and conventional altimetry needed to interpret their ISW sea surface signatures during the Sentinel-6/Jason-3 tandem period.

Tandem calibration stages are common practice in reference altimetry missions and are essential for intercalibration procedures needed in unifying datasets acquired by different instruments [8]. In essence, the newcoming satellite is placed closely in orbit with its predecessor, meaning that during this stage both instruments can be assumed to measure the same ground-truth and any differences therein should be of instrumental (or processing) origin. Nevertheless, Sentinel-6 is the first SAR altimeter to be introduced in the altimetry reference record [7], meaning that the differences between SAR and conventional altimeters need to be accounted for. Note that these differences may be of instrumental nature, but not only. Another source for differences can be attributed to the different acquisition geometries between a conventional and a SAR altimeter. On the one hand, as illustrated in Figure 1, the ground footprint in a conventional altimeter is circular-shaped and typically a few kilometres (depending on sea state conditions). On the other hand, the ground footprint in a SAR altimeter is very different. It is rectangular in shape and sharpened in the along-track spatial resolution (around 300 m for the unfocused SAR in Sentinel-6), whereas in the across-track direction it is still limited to the diameter of a conventional altimeter (for more details see e.g., [13] and references therein).

In our case, that means differences between Sentinel-6 and Jason-3 can still be explored while safely assuming that they are not related to the characteristics of the illuminated surface. But it cannot be neglected that different ocean phenomena will be sensed differently

depending on how their spatial scales and orientation are sampled in the across and along-track acquisition geometries of a SAR versus a conventional altimeter. Interestingly, that may ultimately mean that the assumption of a uniform Brown surface may be realistic in the former but invalid in the latter. This is especially important in ocean phenomena with spatial scales raging in-between those of SAR and conventional ground footprints (i.e., hundreds of meters to kilometres).

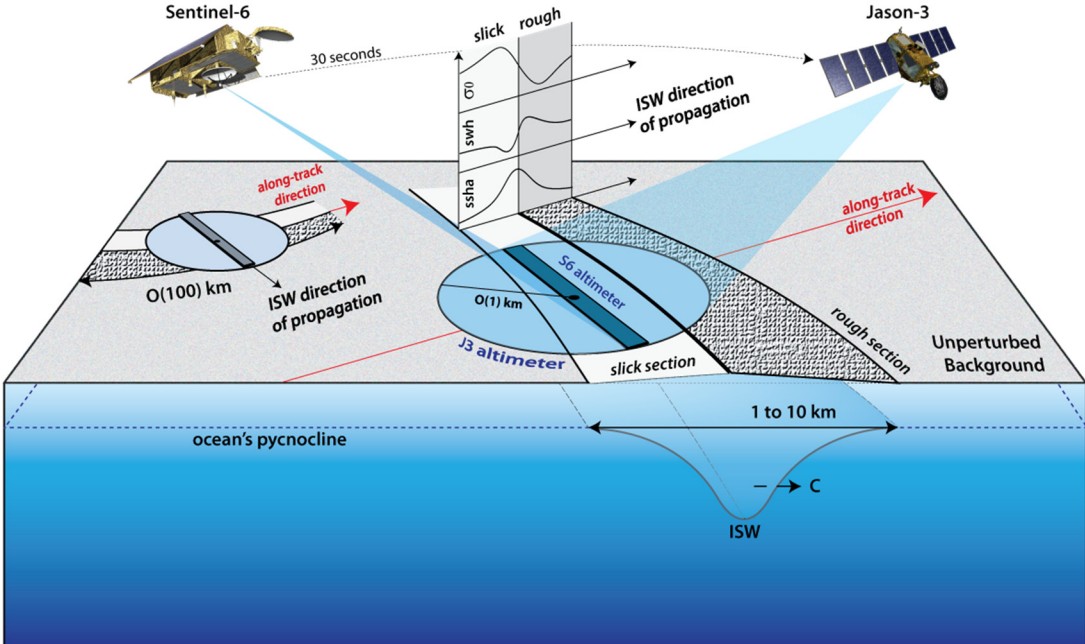

**Figure 1.** Schematic representation of an ISW propagating with phase velocity $\vec{c}$ and its sea surface roughness patterns with leading rough and trailing slick-like sections. Footprints are shown over the slick-like section for Jason-3 (round-shaped) and Sentinel-6 (rectangular)—assumed at nadir but shown at an angle for simplicity. Note that spatial scales are chosen to highlight that, when properly aligned, the smaller footprint in a SAR altimeter can sample consecutive echoes in each section of the ISW, while the larger footprint in a conventional altimeter can obtain mixed contributions from both sections simultaneously. However, both satellites can obtain mixed contributions as illustrated in the alternative view on the back left-hand side.

This issue is the focus of this study, which is investigated from the vantage point of the Sentinel-6/Jason-3 tandem period (from December 2020 to March 2022) and their sea surface measurements of ISWs. Note that with a delay of just 30 s between these satellites the ISWs can be considered stationary (their typical propagation velocities in the ocean are of the order of 1 m/s, see e.g., [17]) and hence their measurements simultaneous in both SAR and conventional altimeters. This is illustrated in Figure 1, which shows that typical scales in ISWs can modulate an unperturbed sea surface into consecutive arrayed sections of increased and decreased sea surface roughness. This inevitably breaks down the assumption of a uniform Brown surface in the transition between their rough and slick-like sections, which can be considered instantaneous at the scales of both SAR and conventional altimeters. At the same time, the rough and slick-like sections of the ISWs are expected to be sampled by a number of individual echoes, which in principle should be different between Sentinel-6 and Jason-3 and hence can be explored to compare how SAR and conventional altimeters perform.

A word of caution is warranted nonetheless regarding the alignment between altimeters and ISWs (see Figure 1). If moving in the same direction a sufficient number of individual waveforms are expected in each section of the ISWs from the sharpened waveforms in the Sentinel-6 altimeter, while mixed contributions can contaminate the larger

footprint in Jason-3. However, in realistic ocean conditions this convenient alignment is likely to be an exception. When propagating at a generic angle with the altimeter, the ISW's rough and slick-like sections can be sampled simultaneously and hence introduce mixed contributions in both SAR and conventional altimeters (which share their across-track resolution)—as illustrated in the back left-hand side of Figure 1 in the limiting case of a right angle between the altimeters' track and the ISWs' propagation direction (see also the recent investigations in [18]).

When properly aligned, Santos–Ferreira et al. [13–15] have shown that SAR altimeters aboard Sentinel-3 can unambiguously detect the sea surface signatures of large-scale ISWs (i.e., with scales of a few kilometres), including in radar backscatter, SSHAs, and SWHs. As illustrated in Figure 1 (see also Figure 4 in [13]), we recall that according to theory, an ISW should have characteristic signatures in these signals. Namely, positive SSHAs are expected centred with the ISW of the order of a few tens of centimetres, together with increasing SWHs and decreasing $\sigma_0$ in its leading (rough) section, and decreasing SWHs and increasing $\sigma_0$ in its trailing (slick-like) section. Similar sea surface signatures can be also found in the SAR altimeter aboard Sentinel-6, which we present and investigate in the next section.

Before proceeding with our results, however, we recall that radar pulses being reflected back to the altimeter (i.e., backscattered) are recorded over a period of time, generating what is commonly referred to as "return waveforms". These waveforms have been investigated in [10,11] in the case of homogeneous, isotropic, and perfectly reflecting surfaces, which showed that waveforms returning from the ocean surface can be processed to fit these idealized conditions in order to retrieve valuable geophysical information—namely, NRCSs, SSHAs and SWHs. This process of fitting is known as *retracking* (see e.g., [8]). This is important because, since Brown [10] and Hayne [11], a series of different retracking algorithms have been developed to accommodate the intrinsic technological developments in the successive altimetry missions, their ability to sense different features of the sea surface, and the different demands therein. This is no different for sea surface signatures of ISWs, and for the purposes of this study, we briefly highlight here a few particularly important retracking algorithms for both conventional and SAR altimeters.

On the one hand, for conventional altimeters, we briefly refer first to the maximum likelihood estimator, MLE3, since it was a pioneer in a series of altimetry applications [19,20], but which nonetheless has been shown to perform poorly in the detection of ISWs (see e.g., [12])—and hence will not be pursued further in this study (except when needed in interpreting our results). Its successor, the MLE4 that is widely used in conventional Jason missions (see e.g., [21,22]), was shown to perform much better in detecting the sea surface manifestations of ISWs [12]. Still in the framework of conventional altimetry, we highlight a third retracking algorithm (ALES, see full details in [4]) that essentially selects the initial stages of the waveforms (i.e., neglects parts of the trailing edges) to improve the altimetry capabilities in near-coastal regions. The last of the algorithms to be investigated in conventional altimetry is the Jason-3 Adaptive retracker. This particular algorithm was shown by [23] and references therein to perform well across a wide range of ocean phenomena, including ISWs. A novelty in this product is that it includes sea surface roughness in the retracking algorithm, including in cases of ocean specular waveforms (see also [20] for more details)—which is what is expected in ISWs as they modulate the sea surface in alternating rough/slick-like sections (see Figure 1). We anticipate that these two retracking algorithms (i.e., ALES and the Adaptive retracker) will be especially important in this study (to be further discussed).

On the other hand, waveforms in SAR altimeters are more complex and need special retracking algorithms (e.g., the SAMOSA model used in Sentinel-3 [24,25]). In level-2 altimetry products, these are labelled as the *Ocean Retracker*, which we use in the following results whenever referring to both Sentinel-6 and (only briefly) Sentinel-3. Finally, as noted in the Introduction Section, we recall that Sentinel-6 acts simultaneously as a SAR (i.e., HR)

and a conventional (i.e., LRM) altimeter, which will also be examined—namely, the ability of the LRM to reproduce the ISW sea surface signatures in Jason-3.

## 3. Results

To investigate ISW sea surface signatures in SAR and conventional altimeters we focus mainly on the Banda Sea in the Indian Ocean. This region has been previously identified as a hotspot for large-scale ISWs [26,27], and used in [15] to document ISW sea surface signatures in the SAR altimeter aboard Sentinel-3. In addition to the Banda Sea, added value case studies are also presented in nearby regions, given that these seas are known hotspots for ISWs (e.g., the Celebes and South China Seas, e.g., [28,29]).

We note in passing, that the data used in these analyses is that provided in standard Level-2 products for both Sentinel-6 and Jason-3 (with the exception of ALES)—given that they provide geophysical data widely used in numerous ocean applications and are easily accessible to the wider oceanographic community (level-1 and raw altimetry data products are not included in this study). This has an important implication in SSHAs in Jason-3 (in the MLE4 retracker) which are only provided at 1 Hz. We further note that while analysing these data, and for a more systematic approach of the several retracking algorithms listed in Section 2, the results that follow focus first on comparing the SAR data in Sentinel-6 with that of Jason-3 provided by the MLE4, ALES and Adaptive retrackers—which constitute the primary focus of this study. We then leave the remaining LRM data in Sentinel-6 to be presented at the end of this Section, since we anticipate that it is consistently observed to be very similar to MLE4 in Jason-3 (as expected).

A typical view of the Banda Sea is presented in Figure 2, where a Synthetic Aperture Radar (SAR) image (acquired 7 March 2022 at 10 h UTC) shows three ISW packets propagating to the north/northeast, which conveniently align with passage 253 in Sentinel-6/Jason-3 (as in Figure 1). We note in passing that SAR images excel in detecting ISWs, whose leading and trailing sections increase (i.e., rough) and decrease (i.e., slick-like) the sea surface roughness and appear as bright and dark parallel bands on the greyish radar background (respectively)—but the reader is referred to [30–32] for the fundamentals of SAR imaging of ISWs. In Figure 2, nonetheless, we stress that ISWs appear mostly as bright bands, whereas the darker slick-like sections of the waves are hard to distinguish in the grey radar clutter. Still in Figure 2, the interpacket distances are observed to be roughly around 140 km, which we note compare well with typical wavelengths for consecutive packets of ISWs generated with a semidiurnal periodicity (M2 = 12.42 h). These two features—the waves' bright signatures and semidiurnal periodicity—will be important in interpreting the altimeter results that follow. In particular, while in their tandem configuration, Sentinel-6 and Jason-3 flew over this region approximately two semidiurnal tidal cycles after the image acquisition in Figure 2 (pass 253, on 8 March 2022 at 13 h 34 m UTC). The records of their level-2 products (for $\sigma_0$, SSHAs and SWHs) are presented in Figure 3 (respectively).

Figure 3a–c begins by showing characteristic sea surface signatures of ISWs in Sentinel-6 (cf. blue envelope around $-6.25°$N), which are very similar to those in Figure 4 in [13] for the SAR altimeter in Sentinel-3, and hence consistent with those expected from two-layer models of ISW theory (as illustrated in our Figure 1, see also [33]). Namely, depressions in $\sigma_0$ (around 1 dB) relative to an unperturbed background are seen with increases in SSHAs and SWHs of about 0.2 m and 1 m, respectively. Note that this is consistent with the view in the SAR image (from Sentinel-1) in Figure 2 since ISWs are mostly seen there as bright stripes, meaning an increase in backscatter in the SAR image (Figure 2) and hence a decrease in $\sigma_0$ in the altimeter in Figure 3a (as illustrated in Figure 2 in [12]). On the other hand, for the same ground truth, the tandem measurements in Jason-3 (in the red lines in Figure 3a–c), resemble much more those of Figure 3 in [12] for a conventional altimeter (in their case Jason-2). While oscillations in radar backscatter of about 2 to 3 dB are still associated with the ISWs, SSHAs (provided at 1 Hz in level-2 products for MLE4) and SWHs seem to provide less detail of the individual waves.

This picture persists for two other representative case studies in the Celebes and South China Seas (shown in Figures 4 and 5, respectively). Again, as expected from theory, the SAR altimeter in Sentinel-6 shows $\sigma_0$ oscillations around 2 to 4 dB, simultaneously with increases in SSHAs and SWHs of about 0.4 m and 2 m, respectively. An exception is noted nonetheless, for the South China Sea (cf. leading ISW around 21.4°N), in which Sentinel-6 shows negative SSHAs and SWHs have excessively high values above 4 m (to be discussed in the next section). At the same time, the corresponding measurements in Jason-3 (using MLE4), also show similar oscillations in $\sigma_0$ (although lower by about 4 dB), but the detail in the waves structures worsens and MLE4 level-2 products are often flagged as missing data (see broken lines in Figures 4 and 5, especially for the South China Sea). Furthermore, estimates in SWHs in the Celebes Sea (Figure 4) appear somewhat unrealistic in the view of ISW theory (e.g., excessively high SWHs around 6 to 10 m).

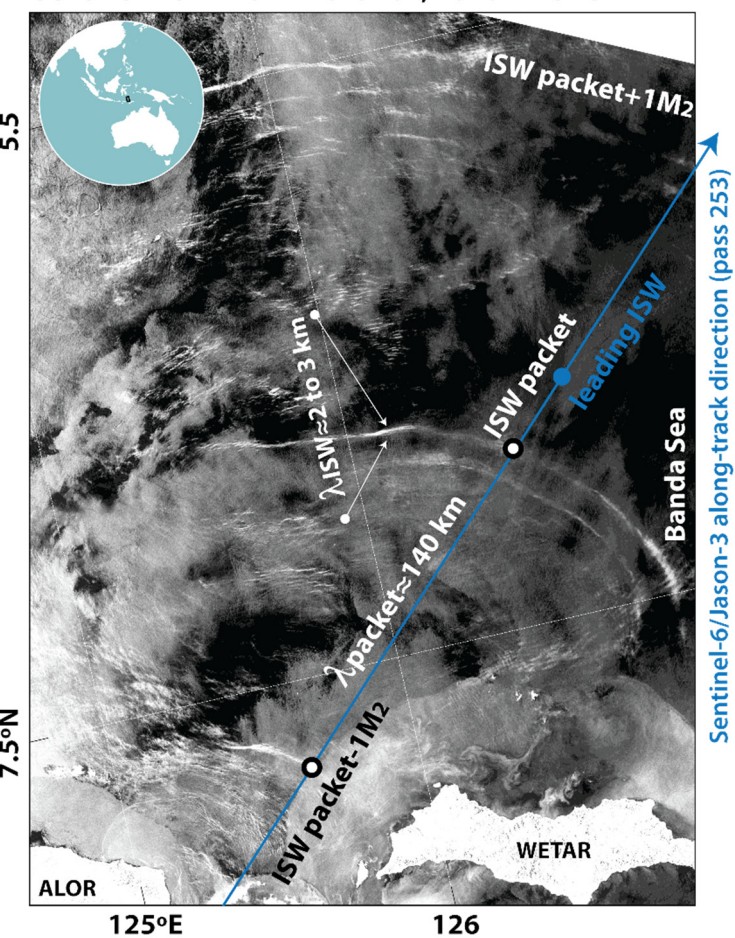

**Figure 2.** Sentinel-1 image over the Banda Sea (see inset in top-left corner) acquired 7 March (2022) at 10h01m UTC. Three ISW packets are seen propagating to the north/northwest separated by typical semidiurnal periods and wavelengths (i.e., from −1M2 to +1M2, with packets separated approximately by $\lambda_{packet} \approx 140$ km). Note that, the width of the leading ISWs in each packet ($\lambda_{ISW}$) is around 2 to 3 km. For reference, the ground-track of Sentinel-6/Jason-3 is shown in a blue line for pass 253 and the blue circle marks the location of the leading ISW-like signals highlighted in Figure 3.

Nonetheless, in these three cases, a new and striking feature is apparent in the tandem view between Sentinel-6 and Jason-3 (using MLE4)—i.e., the radar backscatters between the SAR and the conventional altimeters appear to mirror each other (meaning opposite sea surface signatures in $\sigma_0$). This is a new and puzzling feature that stands out from the vantage point of the Sentinel-6/Jason-3 tandem phase. These opposing radar backscatters

between the SAR and conventional altimeter, which in hindsight may have been present all along in previous studies, should not (in principle) be related to the characteristics of the illuminated target (i.e., within the altimeters' footprints), and hence are worthwhile exploring in more detail.

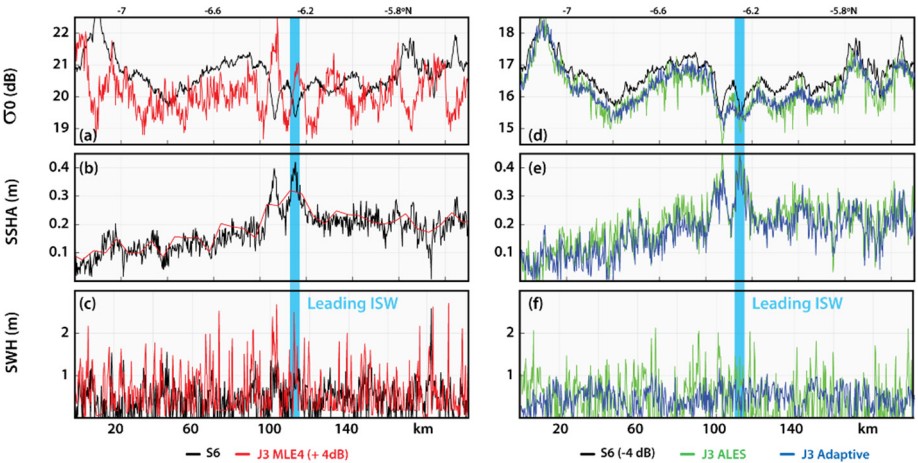

**Figure 3.** Panels (**a**–**c**) show $\sigma_0$, SSHAs and SWHs obtained from Sentinel-6 (S6) and Jason-3 (J3) MLE4 level-2 products (at 20 Hz, except SSHAs for MLE4) in the Banda Sea (8 March 2022, see Supplemental Materials S1). The radar backscatter from Jason-3 (J3) is shown with an offset to highlight its opposite modulations in the vicinities of the ISWs (leading ISW marked with a blue rectangle). Panels (**d**–**f**), same as previous panels for Jason-3 processed with ALES and the Adaptive retrackers (at 20 Hz). Note that panel (**d**) also shows the radar backscatter from Sentinel-6 with an offset to highlight its correlation with the ALES and the Adaptive retrackers.

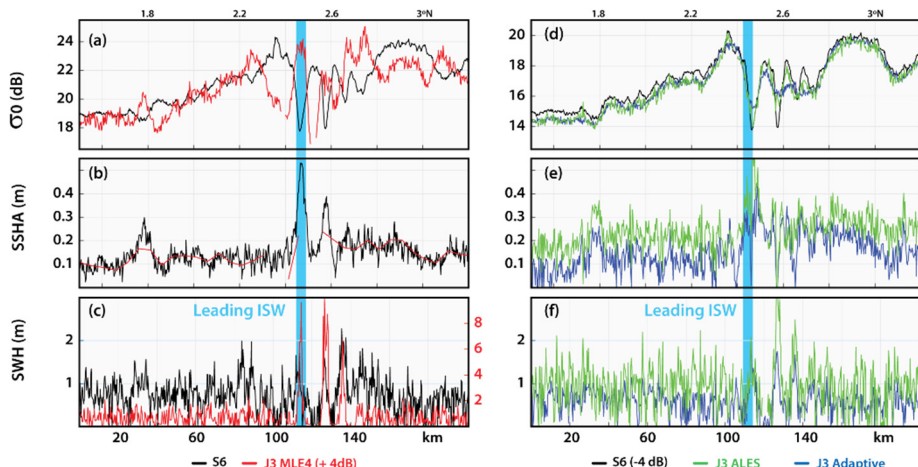

**Figure 4.** Similar to Figure 3 in the Celebes Sea (5 January 2022, see Supplemental Materials S1). Note that in this case, the broken lines in panel (**a**,**b**) indicates missing data in MLE4 level-2 products.

To verify if these cases are isolated or recurring events, a systematic search was performed in pass 253 in the Banda Sea for similar ISW signatures during the Sentinel-6/Jason-3 tandem phase. In this search, possible ISW-like signatures in the altimetry record were supervised independently with other satellite images to ensure the existence of ISWs within approximately two semidiurnal tidal cycles. Four more cases were found with similar evidence as that in Figures 3–5, making seven cases in total, which we summarize in our Supplementary Materials S1 and S2.

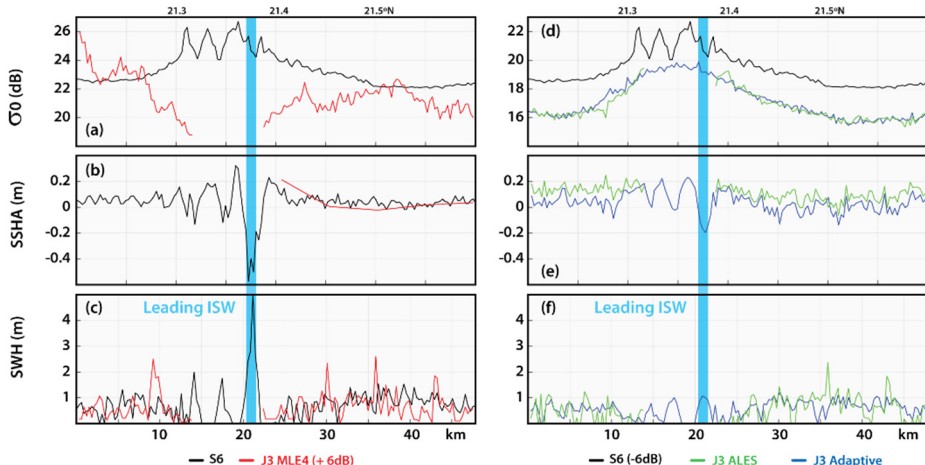

**Figure 5.** Similar to Figure 3 in the South China Sea (15 May 2021, see Supplemental Materials S1). Note that in this case, the broken lines indicate missing data in MLE4 level-2 products in panels (**a–c**) and in the ALES retracker in panels (**e,f**).

We note in passing that, according to Table S1, our supervised cases cover different surface winds and local surface wave fields (i.e., SWHs), which range approximately from 2 to 7 m/s and 0.4 to 1.3 m, respectively), and hence representative of typical open ocean environmental conditions. Additionally, to avoid rain-affected measurements, cases with radiometer liquid water or water vapour above 0.01 g/cm$^2$ or 60 g/cm$^2$, respectively were discarded as being suspicious of rain (just as in [12,13]). We further note that Table S1 provides estimates for ISW propagation velocities when assuming that the observed ISW-like features in the altimeters could be traced back or forward to visual references for the waves in independent satellite imagery (see permalinks in Table S1). These estimates consistently provide propagation velocities around 3 m/s, which compare well with estimates from consecutive ISW wavepackets in satellite imagery, making a stronger case that the features in our supervised cases are indeed those of ISWs—e.g., in Figure 2a interpacket distances of about 140 km translate to propagation velocities around 3.1 m/s if a semidiurnal period is assumed (i.e., 1M2 = 12.42 h).

A simple way to investigate how radar backscatter ($\sigma_0$) compares between SAR and conventional altimeters (i.e., Sentinel-6 and Jason-3) is via Pearson's correlation coefficients (R), which are commonly used to assess how two given variables are related. However, for more meaningful comparisons in our supervised cases, two distinct sections were analysed separately and are listed as follows: one corresponding to ISW-like signals and one other for an undisturbed background (i.e., away from the ISWs and hence unaffected by them). For the first case, to isolate the ISW-signals in Sentinel-6/Jason-3, spatial averaging was completed in the original 20-Hz data using typical length scales for the waves observed in the altimeters (i.e., around the kilometre scale), which in essence removes any higher-frequency content contaminating the ISWs' signals. In the second case, comparisons between Sentinel-6 and Jason-3 were also completed using this spatially smoothed signal, in which the unaffected background was taken immediately ahead (i.e., upstream) of the ISW-like signals. Note that, altogether, this means that in the analyses that follow it is implied that frequencies higher than those in the ISWs are not accounted for in our correlation coefficients—neither in the waves nor in their reference backgrounds. This is illustrated in Figure 6 for the Celebes Sea as a representative example. The correlation coefficients for our supervised cases in Figure 6 (in red circles) confirm that the radar backscatters sensed in Sentinel-6 and Jason-3 (using MLE4) are consistently negatively correlated in the presence of large-scale ISWs (i.e., one mirroring the other as in Figures 3–5 and in Supplemental Materials S2). In the unperturbed background data, however, Pearson's coefficients appear scattered around zero, hence pointing to neither positively nor negatively correlations (to be discussed in the next Section).

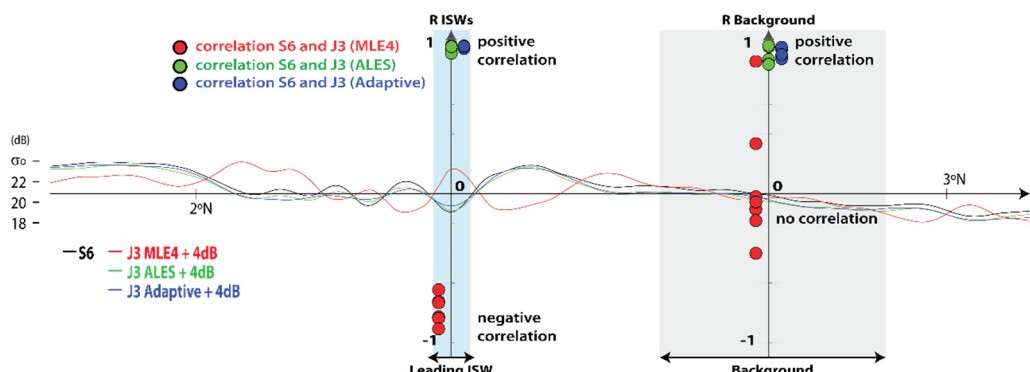

**Figure 6.** Correlations coefficients (*R*) for all cases listed in Supplementary Materials S1 and computed as described in Section 3. Green and blue circles represent correlations between Sentinel-6 and Jason-3 (ALES/Adaptive, respectively), and red circle represent correlations between Sentinel-6 and Jason-3 (MLE4). Two sets of correlation coefficients are shown, listed as follows: on the left for the leading ISWs ($R_{ISWs}$), and on the right for the waves' background conditions ($R_{Background}$). A representative case is show for the Celebes Sea using data smoothed with a running mean of about 10 km (see also Figure 4). Note that correlation coefficients for the South China Sea are only shown for the waves' background (owing to missing data in the ISWs, see Figure 5).

While recalling that these results represent commonly used retracking algorithms provided in level-2 products (MLE4 for Jason-3 and the ocean retracked for Sentinel-6), it can be wondered if they still hold for other retrackers as well. In particular, the ALES and Adaptive retracking algorithms for Jason-3 are especially interesting. This is because ALES (the Adaptive Leading Edge Subwaveform retracker [4]) is meant to deal with sharp land-sea transitions in conventional radar altimeters, making it a natural candidate to perform in the sharp $\sigma_0$ transitions in ISWs. Similarly, the Adaptive retracker is meant to accommodate to various conditions in sea surface roughness, including ocean specular waveforms, which is precisely the case when dealing with these waves' rough and slick-like sections (see Figure 1)—and hence also expected to perform well in ISWs.

Therefore, the radar backscatter from ALES and the Adaptive retracker are also shown for our selected case studies in Figures 3–5 (panels d to f, in green and blue lines, respectively), which show a contrasting picture from our previous results. It can be readily seen that, unlike the MLE4 (in red), these algorithms provide $\sigma_0$ values for the conventional altimeter in Jason-3 that are very similar (but lower in magnitude by about 4 to 8dB) to those in the SAR altimeter in Sentinel-6 (in black). Consequently, it is not surprising that the corresponding correlation coefficients for the ALES/Adaptive retrackers in Figure 6 (in green and blue circles) cluster close together around 1 for both the ISWs and the unperturbed background.

An overall view of these results is provided in Figure 7. Our supervised cases consistently show (except in the South China Sea) modulations in the radar backscatter in the leading ISWs (i.e., $\Delta\sigma_0$) are either opposite or similar between Sentinel-6 and Jason-3, depending on if the latter was processed with either MLE4 or ALES/Adaptive, respectively). In addition, $\Delta\sigma_0$ in Jason-3 with MLE4 is observed to be systematically 1 to 2 dB higher than in Sentinel-6—but it is important to stress that the absolute values in Jason-3 (with MLE4) are 4 to 6 dB lower than in Sentinel-6, both in the ISWs as well as in their preceding background (see Figures 3–5 and Supplementary Materials S2). In the ALES/Adaptive data, $\Delta\sigma_0$ values have magnitudes much closer to those in Sentinel-6, but are also 4 to 8 dB lower in their absolute values.

The changes in SSHAs owing to the leading ISWs (i.e., $\Delta$SSHA) are on average of the order of +10 cm in Sentinel-6 (also except in the leading wave in the South China Sea), whereas in Jason-3 the 1 Hz MLE4 level-2 data degrades the detail provided in the waves' spatial structure (or is simply flagged as missing data). The ALES/Adaptive data, which is

processed at 20 Hz, seem to provide estimates closer to those in Sentinel-6, even though some loss in detail is also found in these retrackers (see also Supplemental Materials S2).

| Supervised Cases | Δσ0 ISW → C S6 | Δσ0 J3MLE4 | Δσ0 J3ALES | Δσ0 J3 Adaptive | ΔSSHA ISW → C S6 | ΔSSHA J3MLE4 | ΔSSHA J3ALES | ΔSSHA J3 Adaptive | ΔSWH ISW → C S6 | ΔSWH J3MLE4 | ΔSWH J3ALES | ΔSWH J3 Adaptive |
|---|---|---|---|---|---|---|---|---|---|---|---|---|
| 2021 Apr. 05 Banda Sea | +0.5/-0.5 | -1/+1 | +0.5/-0.5 | +0.5/-0.5 | +0.2 | +0.1 | +0.2 | +0.1 | 1.3 | 1.8 | 1.4 | 0.5 |
| 2021 Jul. 13 Banda Sea | +0.5/-0.5 | -1/+1 | +0.5/-0.5 | +0.5/-0.5 | +0.1 | +0.1 | +0.1 | +0.1 | 1.1 | 1.4 | 2.0 | 1.0 |
| 2021 Sep. 11 Banda Sea | +0.5/-0.5 | -2.5/+2.5 | +0.5/-0.5 | +0.5/-0.5 | +0.2 | +0.1 | +0.2 | +0.1 | 1.5 | 1.7 | 1.5 | 1.2 |
| 2021 Oct. 10 Banda Sea | +0.5/-1 | -3/+3 | +0.5/-1 | +0.5/-0.5 | +0.1 | +0.2 | +0.1 | +0.2 | 1.1 | 1.5 | 1.4 | 0.9 |
| 2022 Mar. 08 Banda Sea, Fig. 3 | +1/-1 | -2/+2 | +1.5/-1.5 | +0.5/-1 | +0.2 | +0.1 | +0.2 | +0.2 | 1.1 | 2.3 | 1.5 | 0.7 |
| 2022 Jan. 05 Celebes Sea, Fig. 4 | +4/-5 | -7/+5 | +4/-5 | +3/-2 | +0.4 | NA | 0.3 | 0.4 | 1.7 | 9.9 | 2.0 | 1.9 |
| 2021 May 15 S. China Sea, Fig. 5 | +2/-2 | NA | NA | +0/-0 | -0.7 | NA | NA | -0.4 | 5 | NA | NA | 1.0 |

**Figure 7.** Overview of selected cases listed in Table S1 (in Supplementary Materials S1) showing modulations in radar backscatter ($\Delta\sigma_0$), SSHAs ($\Delta$SSHA) and SWHs ($\Delta$SWH) in the leading ISWs. Note that for consistency, all values assume the waves are travelling rightwards (with phase velocity *C*). In the case of $\Delta\sigma_0$, each pair represents the backscatter modulation in the rear/leading sections in comparison with an unperturbed background (taken ahead of the ISW). Yellow circles mark either missing values in level-2 products (NA) or estimates that are not in agreement with two-layer solitary wave theory.

Similarly, SWH oscillations (i.e., $\Delta$SWH) in Sentinel-6 are of the order of 1 m (again, except in the leading wave in the South China Sea), while Jason-3 provides slightly higher estimates for MLE4 (sometimes too high, e.g., in the Celebes Sea case study in Figure 4). Overall, SWHs in ISWs in the ALES/Adaptive retrackers are similar between them and closer to those in S6 (with ALES yielding slightly larges values)—whereas again with some loss in detail in the waves' structure (or none at all as in the ALES data in the South China Sea, see also Supplementary Materials S2).

Finally, we now turn to the LRM data in Sentinel-6, which is presented in Figure 8 for the Banda Sea (cf. also Figure 3) and in Supplementary Materials S3 for the remainder of the case studies listed in Table S1. According to these results, the LRM products in Sentinel-6 (i.e., the equivalent to conventional altimetry) are consistently found in all cases to reproduce very closely the MLE4 data in Jason-3 (but about 1 to 2 dB lower) and hence also in phase opposition with the simultaneous Sentinel-6's HR data (see Figure 8, and Supplementary Materials S3, but note that LMR data are only available in level-2 products after June 2021).

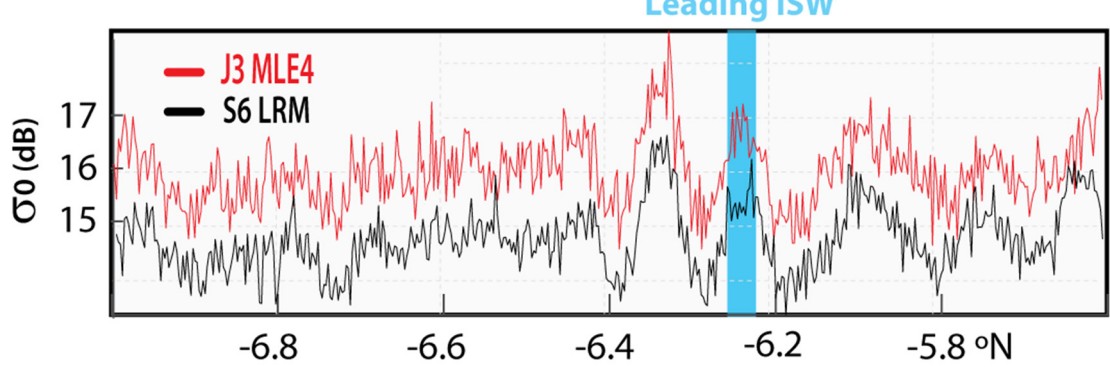

**Figure 8.** Similar to Figure 3a, but comparing $\sigma_0$ for Sentinel-6 (S6) level-2 products (at 20 Hz) in LRM (i.e., conventional altimeter) modes, and data from MLE4 in Jason-3 (J3).

## 4. Discussion

The ISWs measurements from Sentinel-6 presented in this study confirm the capabilities of SAR altimeters to detect these waves' sea surface signatures in level-2 products—namely in radar backscatter (or $\sigma_0$), SSHAs, and SWH—similarly to results already presented for the SAR altimeter in Sentinel-3 [13–15]. However, from the vantage point of the Sentinel-6/Jason-3 tandem phase there are a few issues that warrant further discussions. Following the same structure as in Section 3, we first discuss the results in Figures 3–6, which concern the HR in Sentinel-6 and the MLE4 and ALES/Adaptive data in Jason-3—i.e., the primary focus of this study. We then turn to the LRM data in Sentinel-6 at the end of this section.

The first feature in need of clarification is the apparent opposite phases in $\sigma_0$ between Sentinel-6/Jason-3 in the ISW-like signals when using the MLE4 retracker. In addition, it is also puzzling why these negative correlations between the SAR and conventional altimeters in the ISWs quickly change to positive correlations when the ALES/Adaptive retrackers are used instead. A second issue is also found in the ISWs background, in which data in Sentinel-6 and Jason-3 appear over a wide correlation range in the MLE4 retracker (with most cases being essentially decorrelated), but again well-correlated in the ALES/Adaptive retrackers. Finally, a third issue is seen in the South China Sea, where Sentinel-6 and Jason-3 (with the Adaptive retracker) measurements appear not always to agree with those expected from theory, and Jason-3 measurements in both MLE4 and ALES are flagged in level-2 altimetry products as missing data.

While discussing the first issue, it is important to recall that we are assuming that the same ground truth in the ISW-like signals is being measured by both SAR and conventional altimeters. Note that local swell waves might be an exception, but these are likely averaged out in our kilometre-scale running means. Therefore, it is appealing to conjecture that the $\sigma_0$ opposing signals could be a consequence of these sensors' different acquisition geometries—especially in ground resolution as illustrated in our Figure 1—given that all other factors remain the same. Furthermore, we could assume a priori that Sentinel-6 SAR altimetry data are a rightful representative of our ground-truth—especially since it compares well with what we expected from ISW theory—meaning that measurements from Jason-3 (in the MLE4 retracker) would be reversed.

Interestingly, this initial assumption would be consistent with a similar effect already documented in [34] when comparing performances between MLE3 and MLE4 in Jason-2. According to this study, the key issue is that these two retracking algorithms weight measured waveforms differently when fitting them to an ocean model. In essence, MLE3 is more sensitive to leading than trailing edges in the returned waveforms, while MLE4 does precisely the opposite. This can cause these two signals to have opposite phases in $\sigma_0$– especially when dealing with transitions in backscatter in scales close to those of the altimeter's footprint (much like the results presented here). Note that when considering the acquisition geometries in conventional and SAR altimeters, the processing in MLE3 may cause it to converge to Sentinel-6 SAR altimetry data. The reason for that is simply because MLE3 is more focused on leading edges, which narrows down the timespan of the returned waveforms to its initial stages, and hence artificially (i.e., by means of signal processing) reduces the spatial resolution in the illuminated surface. Consequently, in light of the opposing phases between MLE3 and MLE4 in [34], this means that opposing phases would be expected in our ISW cases between the sharpened SAR data in Sentinel-6 and the conventional MLE4 data in Jason-3.

Furthermore, this interpretation would also be consistent with the results presented in [12], in which quasi-synergetic measurements between Jason-2 and an Envisat-ASAR image had already revealed reversed $\sigma_0$ signals in the conventional altimeter when using the MLE3 and MLE4 retrackers. This is shown in their Figure 3c, which confirms that MLE3 is not so effective in detecting $\sigma_0$ changes owing to ISWs (as described in Section 2), but nonetheless presents oscillations in agreement with what would be expected from two-layer ISW theory. Note that in their Figure 3c, the MLE4 is more sensitive to ISWS, and shows

increases in the radar backscatter in the waves' leading rough sections and decreases in the trailing slick-like sections—but these are opposite of what would be expected. In their study, Magalhaes and da Silva [12] also suggested that inhomogeneities inside the larger conventional footprint could trigger this effect, while noting that footprints can be large enough to include slick sections while still centred at the waves' rough patches and vice versa (i.e., as described in [34]). In turn, that would be consistent with the initial assumption of Sentinel-6 providing accurate representations of the waves' rough and slick-like sections, since its sharper along-track resolution would allow individual waveforms to be sampled distinctively in each section separately.

In light of these considerations, however, why would ALES and the Adaptive retrackers perform better than MLE4 in Jason-3, and in fact, for large-scale ISWs, seemingly as good as the SAR altimeter in Sentinel-6?

According to Passaro et al. [4], the ALES retracker was devised to deal with the sharp sea/land transitions in coastal regions. To do that, ALES targets only the initial stages of the conventional waveforms (i.e., the leading slopes, see Section 3.3 in [4]), which in turn essentially means that it is focusing on a smaller part of the altimeter's footprint. This also means that ALES (via its processing algorithm) is sharpening the altimeter's resolution, and hence would be expected that its results converge towards those of a SAR altimeter. In a way, this is similar to the previous explanations for MLE3 and MLE4, especially since ALES is also not sensitive to waveforms' trailing slopes and hence is not expected to conform to MLE4.

The case in the Adaptive retracker is slightly different since it is not systematically favouring any particular part of the altimeter waveforms—unlike MLE3, MLE4 or ALES. This retracker is designed instead to adapt (i.e., accommodate) to a wide range of sea surface roughness conditions, including those corresponding to specular waveforms. However, ISWs cause the sea surface to change between an unperturbed background and rough/slick-like sections, which are affecting mainly the waveforms leading edges (i.e., in radar backscatter). Our results show that this particular algorithm is sensitive to these sharp changes and hence also focusing in the initial stages of the returned echoes. Again, this means that the Adaptive retracker in Jason-3, in the particular case of ISWs, can sharpen the altimeter's resolution (via processing) in cases of quick transitions in $\sigma_0$, and is therefore able to yield similar ISW modulations as those in ALES in Jason-3 and in the SAR data in Sentinel-6.

It is interesting to note that these two retrackers have very different approaches: one is taking advantage of its capability to retrack the section of interest in the waveform data (ALES) and the other accounting for surface roughness in the waveform modelling, which seems especially important in cases where sea surface roughness changes rapidly (Adaptive). Nonetheless, in the ISW examples investigated in this study, it is noteworthy that their results are very similar.

In our second issue, a similar argument could be made to discuss the apparently correlated/decorrelated signals between Sentinel-6 and Jason-3 in the waves' unperturbed background when using either ALES/Adaptive or MLE4 (respectively) (see Figure 6). Note that correlations between retrackers are being assessed in spatially averaged data that retains the frequency content in typical ISW-like and lower frequencies, while excluding higher frequencies. In the waves' background, therefore, that would mean that Sentinel-6 and Jason-3 (with either ALES/Adaptive or MLE4) would be more or less correlated depending on the amount of high-frequency signals there—i.e., note that in principle they could all be perfectly correlated in a smooth low-frequency background. This is in agreement with our cases in Figures 3 and 4, in which a visual inspection quickly shows that the background large-scale trends (i.e., with scales much larger than the ISWs) agree reasonably well in both altimeters (as would be expected, even for MLE4). Nonetheless, signal modulations with scales close to the ISWs (which are still retained in the smoothed signal) are often seen to modulate the background ahead of waves (see e.g., Figure 3 in the Banda Sea)—probably owing to similar high-frequency phenomena such as oceanic fronts, $\sigma_0$ blooms or surface wind variability at small scales (see e.g., [35–37]). These higher-frequency modulations in the background appear to introduce a decorrelation effect

between the radar backscatter in Sentinel-6 and Jason-3, which is more apparent in the coarser scales of MLE4 but not in the finer scales of the ALES/Adaptive retrackers.

The third issue relating to measurements in the South China Sea (in Figure 5, see also Table S1) can also be interpreted in light of different acquisition geometries in Sentinel-6 and Jason-3. Even though the ISW-like signals in our supervised cases are all in the kilometre-scale, there are two fundamental differences that set the South China Sea case apart. On the one hand, the individual waves in this case have characteristic widths of about 1 kilometre and distances between them are of the same order (see Table S1). On the other hand, the direction of wave propagation seems no longer so conveniently aligned with the altimeters ground-track (in this case the angle between them can be as high as 50°). When considering the acquisition geometries illustrated in Figure 1, this means that the SAR altimeter may no longer be able to sample the individual waves distinctively in their rough and slick-like sections. Because the waves are smaller, closer together, and propagating at an angle with the altimeters, the coarser across-track resolution in Sentinel-6 (which is also in the kilometre scale) is more likely to illuminate rough and slick-like sections simultaneously. This introduces conflicting backscatter contributions along the different stages of the SAR waveforms, which could render their retracking poorer and any estimates thereof unrealistic (e.g., as in Figure 5 for SSHAs and SWHs). The same applies in both MLE4 and ALES/Adaptive in the case of Jason-3, whereas in the footprint of a conventional altimeter multiple waves could be sampled at once—meaning even more random contributions from high and low backscattering that ultimately may flag level-2 products as missing data (as in Figure 5d–f).

Finally, we now turn to discussions concerning the results presented in Figure 8 and Supplemental Materials S3, which present LRM data products in Sentinel-6. In these cases, however, our analysis is more straightforward. The LRM data in Sentinel-6 is found in all cases (when data are available) to follow closely those of MLE4 in Jason-3 in both ISW-like signals as well as in the unperturbed background (i.e., in both high and low-frequency signals), which is somewhat expected given that it was designed to provide an equivalent for conventional altimetry. Nonetheless, it is noteworthy that it conforms to MLE4 level-2 data but not to the other retrackers covered in this study (i.e., the ALES and Adaptive retrackers in Jason-3).

## 5. Summary and Concluding Remarks

This study explores simultaneous measurements of ISWs between a SAR and a conventional altimeter from the vantage point of the Sentinel-6/Jason-3 tandem phase. The SAR altimeter in Sentinel-6 shows detailed sea surface signatures of large-scale (i.e., in the kilometre-scale) ISWs that are consistent with those presented previously for the SAR altimeter in Sentinel-3 [13–15]. In general, the results presented here are in fair agreement with those expected from two-layer ISW theory whether in radar backscatter or in SSHAs and SWHs. An exception is highlighted in a case in the South China Sea, where the ISWs' horizontal scales become of the order of 1 kilometre and are observed to propagate at an angle with the altimeter ground-track, for which level-2 products in the SAR altimeter depart from those expected from theory.

The same ground truth in Jason-3 is primarily investigated in light of three retracking algorithms: the conventional MLE4 that is commonly provided in level-2 products, and ALES and the Adaptive retrackers that are originally devised to deal with sharp changes in backscatter (owing e.g., to land/sea transitions). All retrackers show a general loss of detail in the waves' structure in $\sigma_0$, SSHAs and SWHs. However, it is important to recall that SSHAs in MLE4 are only provided at 1 Hz, and hence in this regard a recommendation from this study is that Jason-class level-2 products provide it also at 20 Hz (similarly e.g., to $\sigma_0$ or SWH).

One other retracker is also briefly explored. The LRM product in Sentinel-6 is shown to replicate the MLE4 data in Jason-3 in our ISW case studies, which confirm this heritage mode to replicate conventional altimetry level-2 products when dealing with sharp transi-

tions in sea surface roughness of similar spatial scales as those in conventional altimeters (i.e., of a few kilometres).

Perhaps the most interesting novelty in this study is that in the radar backscatter a new puzzling feature stands out from the synergy between Sentinel-6 and Jason-3. It is consistently shown that these SAR and conventional altimeters are either negatively or positively correlated in ISW-like signals, when using the standard MLE4 or the alternative ALES/Adaptive retrackers in Jason-3 (respectively). These results are discussed in light of the intrinsically different acquisition geometries between SAR and conventional altimeters. It is argued that, when the waves' propagation aligns with the altimeter, the sharper along-track resolution in Sentinel-6 (of about 300 m) can sample the details of ISWs structure. However, the larger footprint in the conventional Jason-3 (typically a few kilometres wide) cannot resolve the same level of detail and eventually is contaminated with conflicting contributions between distinct sections of the same ISW or even multiple ISWs altogether. Note that, the same applies when ISWs propagate at an angle with a SAR altimeter, since the across-track resolution (which is still a few kilometres wide) can also illuminate either distinct sections of the same wave or a series of them at the same time.

Altogether, the evidence provided in the ISW cases illustrates quite well the need of ensuring a uniform Brown surface in the ocean (i.e., a homogeneous isotropic perfectly reflecting surface [10]). When fulfilled, satellite altimetry performs as expected (e.g., in agreement with ISW theory). However, when mixed contributions compete in the same echo (e.g., from an ISW rough and slick-like sections), retrackers seem no longer able to fit the measured waveform into theoretical expectations from ISW theory.

This is important in the wider scope of satellite altimetry, since the ocean surface is far less uniform in the finer scales that SAR altimetry is now pursuing when compared with the traditional larger scales of conventional altimetry. ISWs demonstrate that in this study, whereas other ocean phenomena can yield similar results—namely, fronts, variable surface wind, and other small-scale phenomena that can potentially affect the kilometre-scale altimetry measurements. It can then be wondered how past, present, and future altimetry measurements—with intrinsically different acquisition geometries—can be reconciled in the spatially shorter and temporal higher frequencies of the ocean spectrum. To this end, it is suggested here that alternative algorithms such as the ALES and the Adaptive retrackers, which explore more focused views of the conventional waveforms (namely in the leading edge), can perform in the same level of SAR altimeters when dealing with sharp transitions in ocean radar backscatter (at least down to the kilometre scale).

Nevertheless, the findings presented in this study warrant further investigations, for instance in smaller-scale ISWs or those propagating at larger angles with the altimeters' ground-tracks (as recently investigated in [18] for the SAR altimeter aboard Sentinel-3), or even other smaller-scale phenomena than ISWs. Whichever the case, however, the tandem phase between Sentinel-6 and Jason-3 (likely to be one of a kind) seems to hold the key to reconciling an upcoming era of SAR altimeters with a climatological record of conventional altimetry.

**Supplementary Materials:** The following supporting information can be downloaded at: https://www.mdpi.com/article/10.3390/rs15020392/s1. The following supporting information is provided. Table S1: List of supervised cases showing evidence of large-scale ISWs in both Jason-3 and Sentinel-6; Supplemental Materials S2: Similar to Figure 3 in the paper but for all case studies listed in Supplementary Materials S1; Supplementary Materials S3: Similar to Figure 8 in the paper but for all case studies listed in Supplementary Materials S1 (but note that LMR data are only available in level-2 products after June 2021).

**Author Contributions:** All authors contributed in conceiving and designing the analysis; in collecting/processing the data; in data analysis; and in some stage in writing the paper. All authors have read and agreed to the published version of the manuscript.

**Funding:** Jorge M. Magalhaes is supported by FCT—Portuguese Foundation for Science and Technology under contracts UIDB/04423/2020 and UIDP/04423/2020. This study was funded by EU and ESA, under ESA Contract No. 4000134346/21/NL/AD "Sentinel-6 Michael Freilich and Jason-3 tandem Flight Exploitation (S6-JTEX) study" between the University of Porto and Collecte Localisation Satellites.

**Data Availability Statement:** The data presented in this study can be provided on request, but are openly available in Copernicus Open Access Hub: https://scihub.copernicus.eu/dhus/#/home, (accessed on 8 December 2022); https://eoportal.eumetsat.int/, (accessed on 8 December 2022); and https://www.ncei.noaa.gov/, (accessed on 8 December 2022).

**Acknowledgments:** This work was funded by EU and ESA, under ESA Contract No. 4000134346/21/NL/AD "Sentinel-6 Michael Freilich and Jason-3 tandem Flight Exploitation (S6-JTEX) study" between the University of Porto and Collecte Localisation Satellites. J.C.B. da Silva thanks the Portuguese Fundação para a Ciência e Tecnologia (FCT) under project UIDB/04683/2020. A.M.S.-F. gratefully acknowledges FCT and the UE for a Ph.D. grant SFRH/BD/143443/2019. J.M. Magalhaes thanks the FCT under projects UIDB/04423/2020 and UIDP/04423/2020. I. G. Lapa gratefully thanks the project CC0—204170—ESA/CLS Multisensor for a Master's Degree grant.

**Conflicts of Interest:** The authors declare no conflict of interest.

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
