# Peer review of "Using a Tandem Flight Configuration between Sentinel-6 and Jason-3 to Compare SAR and Conventional Altimeters in Sea Surface Signatures of Internal Solitary Waves"

_remotesensing, doi:10.3390/rs15020392_

Round 1
Reviewer 1 Report
This work is based on observations taken by Sentinel-6 and Jason-3 in their tandem period, focusing on the phenomenon of the internal solitary waves (ISW).
The overall perception is good. The paper is easy to read and its storyline is well clear. It seems to me also sufficiently referenced.
The main feature observed by the authors that worths discussion is the phase opposition between radar backscatters obtained by S6 and J3 (applying the MLE4 retracker), mentioned in section 3 “Results” at lines 323-331 and then discussed in section 4.
The conjecture regarding this issue, explained by the authors in section 4, is interesting even if it doesn’t fully convince me. But, everything is explained with good clarity, thus I recommend acceptance after very few minor revisions, with the objective to share this discussion and get feedback by the altimetry community.
My (minor) specific comments follow.
- At line 114, I suggest to speak in terms of “homogeneous roughness”, instead of “uniform Brown surface”. This is because the Brown waveshape characterizes the returned radar signal after a single pulse and not strictly the surface. Speaking about surface homogeneity looks more proper, especially when dealing with the SAR processing.
- The long sentence between lines 162-166 should be rephrased.
- At line 185 it’s maybe better to say “…propagating at a generic angle…” and I suggest the Authors to rephrase the whole sentence between 181-190, specifying that, in general, mixed contributions have to be expected at any generic angle between ISW propagation and satellite orbit directions. Perfect alignment or orthogonal directions can be considered exceptional (or “extreme”) conditions.
Author Response
The authors are very grateful for the reviewer’s careful reading of this paper as well as comments and corrections. Detailed responses to each concern may be found below (reviewer’s comments in italics).    
Reviewer #1:
This work is based on observations taken by Sentinel-6 and Jason-3 in their tandem period, focusing on the phenomenon of the internal solitary waves (ISW). The overall perception is good. The paper is easy to read and its storyline is well clear. It seems to me also sufficiently referenced. The main feature observed by the authors that worths discussion is the phase opposition between radar backscatters obtained by S6 and J3 (applying the MLE4 retracker), mentioned in section 3 “Results” at lines 323-331 and then discussed in section 4. The conjecture regarding this issue, explained by the authors in section 4, is interesting even if it doesn’t fully convince me. But, everything is explained with good clarity, thus I recommend acceptance after very few minor revisions, with the objective to share this discussion and get feedback by the altimetry community.
R: We thank the reviewer for these encouraging remarks.
- At line 114, I suggest to speak in terms of “homogeneous roughness”, instead of “uniform Brown surface”. This is because the Brown waveshape characterizes the returned radar signal after a single pulse and not strictly the surface. Speaking about surface homogeneity looks more proper, especially when dealing with the SAR processing.
R: Thank you for this note. We agree that it is clearer to refer to homogeneous surface roughness. Nonetheless a reference to a uniform Brown surface is kept in brackets for consistency with other important notes in the Introduction/Discussions.
- The long sentence between lines 162-166 should be rephrased.
R: Done as requested.
- At line 185 it’s maybe better to say “…propagating at a generic angle…” and I suggest the Authors to rephrase the whole sentence between 181-190, specifying that, in general, mixed contributions have to be expected at any generic angle between ISW propagation and satellite orbit directions. Perfect alignment or orthogonal directions can be considered exceptional (or “extreme”) conditions.
R: We appreciate this note, since, in realistic oceanic conditions ISWs and altimeters will indeed be found to propagate within a wide range of possible angles. The full paragraph now reads (new text in red):
A word of caution is warranted nonetheless regarding the alignment between altimeters and ISWs (see Figure 1). If moving in the same direction a sufficient number of individual waveforms is expected in each section of the ISWs from the sharpened waveforms in the Sentinel-6 altimeter, while mixed contributions can contaminate the larger footprint in Jason-3. However, in realistic ocean conditions this convenient alignment is likely to be an exception. When propagating at a generic angle with the altimeter, the ISW’s rough and slick-like sections can be sampled simultaneously and hence introduce mixed contributions in both SAR and conventional altimeters (which share their across-track resolution) – as illustrated in the back left-hand side of Figure 1 in the limiting case of a right-angle between the altimeters’ track and the ISWs’ propagation direction (see also the recent investigations in [18]).

Reviewer 2 Report
The article takes advantage of the simultaneous observations of S6 and J3 (the two sensors are only 30s apart) to observe ISWs and presents a case study. The reasons for the different observation results are analyzed not only in terms of the observation geometry of the SAR/conventional altimeter; but also in terms of the differences between the different retracking algorithms themselves (MLE3, MLE4, ALES, adaptive retrackers, HR).
The study on ISW was done soon after the launch of S6. First of all, I would like to acknowledge the authors' enthusiasm for scientific research; in addition, the study itself is interesting and somewhat innovative. However, I have some questions that need to be explained by the authors.
1. Line488-489:For missing data after retracing, the authors need to explain what kind of calculations would be recognized as missing. For example, reasons such as: values are too large, calculations cannot be performed, etc.
2. Line581-600:I don't think the authors explain the cause of the SWH anomaly in enough detail. Although the altimeter is no longer able to sample individual waves uniquely in their rough and smooth parts when there is an angle between the propagation direction of the ISW and the altimeter track, the retracking results are in the right direction (it will not be -6/10m). So when there is an angle of entrainment, the altimeter will observe rough strips for a longer period of time, which may be the cause of the ever larger SWH. I hope the authors can discuss this aspect.
3. In fact, no matter how the ISW modulates the altimeter echo, it is ultimately the rough/smooth sea surface that modulates the waveform, so the trailing edge of the waveform is much less affected by the ISW than the leading edge. This is why the calculation results of the retracking method focusing on the leading edge are more realistic. Then proposing a new retracking algorithm for the effect of ISW seems feasible at this point (or modifying it from the existing algorithm), and we are looking forward to whether the authors can discuss how to go about correcting for these anomalous SWHs and the implications of doing so.
4. Line610:Should be changed to ‘5. Summary and Concluding Remarks’
5. Appendix S1,S2,S3 not found in the article
6. Line756: Reference 23 should be written in full
